# BR109, a Novel Fully Humanized T-Cell-Engaging Bispecific Antibody with GPRC5D Binding, Has Potent Antitumor Activities in Multiple Myeloma

**DOI:** 10.3390/cancers15245774

**Published:** 2023-12-09

**Authors:** Ying Liu, Ya-Qiong Zhou, Lei Nie, Shan-Shan Zhu, Na Li, Zhen-Hua Wu, Qi Wang, Jian Qi, Bing-Yuan Wu, Shu-Qing Chen, Hai-Bin Wang

**Affiliations:** 1College of Pharmaceutical Sciences, Zhejiang University, Hangzhou 310058, China; ying.liu01@bioraypharm.com; 2Bioray Biopharmaceutical Co., Ltd., Taizhou 318000, China; yqzhou@bioraypharm.com (Y.-Q.Z.); lei.nie@bioraypharm.com (L.N.); shanshan.zhu@bioraypharm.com (S.-S.Z.); lina@bioraypharm.com (N.L.); zhenhua.wu@bioraypharm.com (Z.-H.W.); qi.wang@bioraypharm.com (Q.W.); jian.qi@bioraypharm.com (J.Q.); bingyuan.wu@bioraypharm.com (B.-Y.W.); 3Hisun Biopharmaceutical Co., Ltd., Hangzhou 311404, China

**Keywords:** multiple myeloma, GPRC5D, T-cell-engaging bispecific antibody, BR109

## Abstract

**Simple Summary:**

In recent years, BCMA-targeted therapies have significantly improved the limited efficacy of conventional therapies for multiple myeloma (MM), yet many patients still relapse. GPRC5D, an MM-specific target that is independent of BCMA expression, has emerged as a potential therapeutic intervention for MM. Additionally, T-cell-mediated therapies have shown promise in treating MM. Consequently, we have developed and analyzed the properties of BR109, an anti-GPRC5D × anti-CD3 T-cell-engaging bispecific antibody (TCB). Both in vitro cell assays and xenograft tumor models in mice showed that BR109 has high antitumor efficacy, supporting its clinical progress as a promising approach to multiple myeloma therapy.

**Abstract:**

At present, multiple myeloma (MM) is still an essentially incurable hematologic malignancy. Although BCMA-targeted therapies have achieved remarkable results, BCMA levels were found to be downregulated in patients with MM who relapsed after these treatments. Therefore, the search for other antigens specific to MM has become a priority. Independently of BCMA expression, G-protein-coupled receptor family C group 5 member D (GPRC5D) is mainly expressed in the plasma cells of MM patients, while it is expressed in a limited number of normal tissues. Combining MM-specific antigen GPRC5D and T-cell-mediated therapies would be a promising therapeutic strategy for MM. Recently, we constructed a new anti-GPRC5D × anti-CD3 T-cell-engaging bispecific antibody (TCB), BR109, which was capable of binding to human GPRC5D and human CD3ε. Moreover, BR109 was proven to have relatively good stability and antitumor activity. BR109 could specifically trigger T-cell-mediated cytotoxicity against many GPRC5D-positive MM cells in vitro. Meanwhile, antitumor activity was demonstrated in MM cell line xenograft mouse models with human immune cell reconstitution. These preclinical studies have formed a solid foundation for the evaluation of MM treatment efficacy in clinical trials.

## 1. Introduction 

Multiple myeloma (MM), also known as plasma cell myeloma or simply myeloma, is a cancer of plasma cells, a type of white blood cell that normally produces antibodies [1]. The four most common signs of MM are often referred to as CRAB, an acronym for high calcium, renal failure, anemia, and bone pain. Having these four signs may also indicate that treatment should begin immediately. MM usually leads to extensive osteolytic lesions, which are frequently observed among elderly patients [2]. MM is the second most common hematologic malignancy, and its incidence is elevated in the aging population [3]. The current standard therapies for MM, such as proteasome inhibitors, immunomodulatory drugs, and anti-CD38 targeting drugs, have limited effectiveness [4], but newer therapies, including B cell maturation antigen (BCMA)-targeted antibody–drug conjugates (ADCs) [5,6], T-cell-redirecting bispecific antibodies (TCBs) [7,8], and chimeric antigen receptor T cell (CAR-T) therapies [9,10,11], have shown encouraging results in recent years [12]. Although BCMA is expressed in most malignant plasma cells, its expression is heterogeneous. It has been reported that patients with MM who relapsed after BCMA-targeted therapies have shown downregulation of BCMA [13]. It is therefore vital to explore new treatment methods and identify specific targets to overcome the unmet needs of MM therapies.

G-protein-coupled receptor family C group 5 member D (GPRC5D) is an orphan receptor and a seven-pass transmembrane protein that was first identified in 2001 [14]. GPRC5D is expressed in cells with the plasma cell phenotype, and in the primary MM cells in particular. Among normal tissues in non-plasma cells, GPRC5D is predominantly expressed in hair follicles and skin [15,16,17]. The mRNA levels of GPRC5D are higher in MM plasma cells than in normal cells (as determined in samples of various tissues from healthy donors), as well as in cells from other hematologic malignancies (e.g., AML, CML, DLBCL, FL, MCL, MGUS, SMM, etc.), which suggests that GPRC5D might represent a potential target for effector-cell-mediated MM therapy [18,19]. It has been suggested that, in 65% of MM patients, detectable GPRC5D has an expression threshold in MM cells of more than 50%; this expression threshold is independent of the detectable BCMA [15]. Therefore, GPRC5D can be used as a therapeutic target to treat MM in BCMA-negative patients. Other reports indicate that the expression levels of GPRC5D are intimately associated with the overall survival of MM patients; therefore, GPRC5D could serve as a prognostic biomarker for MM [20]. Up until now, based on the statistics taken from public data, the GPRC5D-targeted therapies that are currently used clinically include TCBs, CAR-T, and ADC therapies [21,22]. For example, talquetamab (TCB) [23], RG6234 (TCB) [24], BMS-986393 (CAR-T) [25], OriCAR-017 (CAR-T) [26], and LM-305 (ADC) [27] have demonstrated good clinical outcomes and tolerable safety profiles. As a TCB, talquetamab can be administered directly subcutaneously upon purchase, making it convenient and faster than CAR-T therapies [23]. For another GPRC5D-specific CAR-T, MCARH109, it was reported that the highest dose was associated with two cases of late-onset cerebellar toxicity, which might be related to the low-level expression of GPRC5D in the inferior olivary nucleus of the brainstem [21]. Bispecific antibodies similar to IgG are unlikely to cross the blood–brain barrier; therefore, neurologic adverse events are generally rare and manageable [28]. Thus, TCBs represent a promising tool for cancer treatment in contemporary medicine.

In this study, we focused on the discovery of a new anti-GPRC5D antibody and the construction of an anti-GPRC5D × anti-CD3 TCB. Using both in vitro MM cells and in vivo MM cell xenograft mouse models, the anti-MM activity could be comprehensively evaluated.

## 2. Materials and Methods

### 2.1. Cell Lines and Cell Culture

#### 2.1.1. Cell Lines

The cell lines overexpressing *GPRC5*-family-related genes were constructed by Kyinno Biotechnology Co., Ltd. (Beijing, China), and the details are as follows:Chinese hamster ovary cell line K1 expressing exogenous human *GPRC5D* gene (CHOK1-hGPRC5D);HEK293T cell line expressing exogenous human *GPRC5D* gene (HEK293T_hGPRC5D);HEK293T cell line expressing exogenous cynomolgus monkey *GPRC5D* gene (HEK293T_cynoGPRC5D);HEK293T cell line expressing exogenous mouse *Gprc5d* gene (HEK293T_mGPRC5D);HEK293T cell line expressing exogenous human *GPRC5A* gene (HEK293T_hGPRC5A);HEK293T cell line expressing exogenous human *GPRC5B* gene (HEK293T_hGPRC5B);HEK293T cell line expressing exogenous human *GPRC5C* gene (HEK293T_hGPRC5C).

An effector cell line, called 4E2, is a Jurkat/NFAT-*Luc* cell line that can stably express luciferase genes driven by the NFAT response element, which demonstrated significant expression of human *CD3* genes. MM cell lines MM.1R, MM.1S, and RPMI 8226 were obtained from the Shanghai Cell Bank of the Chinese Academy of Sciences, and NCI-H929 was obtained from ATCC.

#### 2.1.2. Cell Culture

The HEK293T cell line was cultured in DMEM with 10% FBS and 1 μg/mL puromycin. CHOK1-hGPRC5D was cultured in F12K with 10% FBS and 10 μg/mL puromycin. MM.1R and MM.1S were cultured in RPMI1640 medium with 10% fetal bovine serum (FBS), 1 nM sodium pyruvate, and 1% NEAA (100×). RPMI 8226 and NCI-H929 were cultured in RPMI1640 medium with 10% FBS and 0.05 mM β-ME. A T cell activation medium was prepared, containing RPMI 1640 and 10% FBS, and a TDCC medium was prepared, containing RPMI1640, Glutamax, 2% FBS, 1 nM Sodium pyruvate, and 1% NEAA (100×). All the reagents were obtained from Gibco (Thermo Fisher Scientific, Inc., Waltham, MA, USA).

### 2.2. Generation of GPRC5D-Specific mAb

The immunizations were performed in 2 sets of experiments with 50 humanized mice in each, with 25 mice receiving DNA immunizations and 25 mice receiving cell immunizations. The vectors in the DNA immunizations were cloned with inserts encoding human *GPRC5D*, and the cell immunization used CHO-K1_hGPRC5D. Antibody titers were assessed with serum antibody assays, and the spleen cells of mice with the highest antibody titer were fused with SP20 myeloma cells to generate hybridomas. Antibodies that can specifically recognize GPRC5D were selected by a binding assay using HEK293T cell lines, which were stably transfected with human *GPRC5D* (hGPRC5D)/cynomolgus monkey *GPRC5D*, (cynoGPRC5D)/mouse *Gprc5d* (mGPRC5D), or human *GPRC5A/GPRC5B/GPRC5C*. 

After extracting the total RNA of the selected cell line, reverse transcription was conducted to produce the cDNA template. The VL and VH fragments were obtained by carrying out the PCR procedure with subtype-specific primers. The variable region was combined with the human IgG1 constant region to generate a full-length humanized IgG1 mAb. Purifications were performed using affinity chromatography on protein A sepharose. 

### 2.3. Binding Flow Cytometry Assay

Different cell lines were incubated at 4 °C for 30 min with the antibodies, which were diluted in gradients. Afterwards, the cells were washed twice with PBS and collected by centrifugation (5 min, 1200 rpm, 4 °C). Next, the cells were incubated with the secondary antibody (30 min, 4 °C). Following two additional washes, the cells were analyzed by flow cytometry assay. A four-parameter logistic regression was performed to fit the concentration–mean fluorescence intensity (MFI) curve using Prism7 (GraphPad Software). In the test, positive control monoclonal antibodies (mAbs) BM01 and BM02 were set; the variable regions were both derived from the GCDB72 antibody (i.e., talquetamab, a kind of anti-GPRC5D × anti-CD3 TCB from J&J, New Brunswick, NJ, USA) disclosed from patent WO 2018017786 A2. They contained the GPRC5D-binding part of GCDB72, and the constant regions were mouse IgG1 (in BM01) and human IgG1 (in BM02).

### 2.4. Epitope Binning of Anti-GPRC5D Antibody

The epitope binning of the generated anti-GPRC5D mAbs was carried out by competition-binding ELISA. Briefly, HEK293T_hGPRC5D cells were seeded into 96-well plates at 2 × 10^5^ cells/well; then, the tested antibody and the antibody conjugated to Alexa488 were successively added into the holes with the confirmed saturation concentrations. Only the Alexa488 antibody was added to the blank control. The mixture was incubated for 1 h at 4 °C. The cells were then washed in PBS and analyzed using flow cytometry. The MFI of each experimental group/the MFI of each blank control group × 100% was the homology between the fluorescent antibody and the naked antibody. The higher the value, the stronger the homology. Conversely, if the value was lower, the homology was weaker. 

### 2.5. Preparation and Characterization of T-Cell-Engaging Bispecific Antibody

An scFv-Fab-Fc structure was used to construct an anti-GPRC5D × anti-CD3 TCB. The constructed candidate molecule was named BR109. In detail, the Fab arm was the anti-GPRC5D antibody (80A8), the scFv was the anti-hCD3ε antibody, and the Fc was a human IgG1 structure with an L234A/L235A mutation (LALA) and a knob-into-hole mutation (KiH). In the test, positive control TCB BM03 was prepared; it had the same variable region sequence as talquetamab and the same scFv-Fab-Fc structure as BR109. We also constructed another positive control TCB with an IgG-like structure via in vitro assembly; this positive control was called talquetamab analog and had the same sequence as talquetamab. Meanwhile, negative control IsotypeCD3 was a non-targeting TCB with the same structure as BR109 that only binds to CD3 and not to GPRC5D. All the TCBs were purified by affinity chromatography on protein A sepharose and molecular sieves prior to use. The binding of TCB BR109, BM03, and IsotypeCD3 to HEK293T_hGPRC5D cells was analyzed by the flow cytometry assay mentioned above. In order to evaluate the stability of BR109, we performed several treatments, such as high temperature (HT) treatments and a series of pH treatments, and the binding abilities of BR109 to bind to the HEK293T_hGPRC5D cells and Jurkat cells were checked through the flow cytometry assay.

### 2.6. T Cell Activation and T-Cell-Mediated Cytotoxicity

Based on previous research from our laboratory [29], MM.1R cells and Jurkat/NFAT-*Luc* cell 4E2 were co-incubated to evaluate the ability of TCB BR109 to induce CD3-mediated T cell activation when simultaneously binding to both human CD3 and human GPRC5D. The MM.1R cells and 4E2 cells were harvested while in a logarithmic growth phase and then seeded into 96-well plates at 2 × 10^4^ cells/well. The controls were inoculated with the same amount of 4E2 cells and blank medium. Serial TCB concentration gradients were added to the appropriate wells for 6h at 37 °C and 5% CO_2_. The extent of the T cell activation was measured by the Bio-Lite Luciferase Assay System (Vazyme, CAT#DD1201-01). In brief, the 96-well plates containing the cells were removed from the incubator and equilibrated to room temperature for 30 min. An equal volume of Bio-Lite Reagent was added to each well, followed by incubation in the dark for 5 min. The relative light unit (RIU) value was measured at 450 nm with a microplate reader (Spectramax M5; Molecular Devices, LLC., San Jose, CA, USA), and the fold induction was calculated using the formula below:$$Fold\;induction=\frac{RLU\left(sample\right)-RLU\;(backgroud)}{RLU\left(negative\;control\right)-RLU\;(backgroud)}$$

A four-parameter logistic regression was performed to fit the concentration–mean fold of induction curve using Prism7 (GraphPad Software).

By using human peripheral blood mononuclear cells (PBMCs, donor ID: LP221213007, ORICELLS, Suzhou, China) as effector cells and MM cell lines MM.1R, MM.1S, and RPMI 8226 as target cells, the LDH assay was used to assess T-cell-dependent cell cytotoxicity (TDCC). We seeded 10,000 MM cells and 100,000 human PBMCs into 96-well U-bottom plates and incubated them for 24 h at 37 °C and 5% CO_2_ with serial TCB concentration gradients. The individual experimental settings are summarized as follows:To each well, 120 μL TDCC medium was added (blank group);To each well, 80 μL TDCC medium and 40 μL MM cells were added (minimum release);To each well, 80 μL TDCC medium and 40 μL MM cells were added (maximum release);To each well, 40 μL TDCC medium, 40 μL PBMCs, and 40 μL MM cells were added (background group);To each well, 40 μL TCBs, 40μL PBMCs, and 40 μL MM cells were added (experimental release).

Target cell killing was measured by an LDH Cytotoxicity Detection Kit (Roche, Indianapolis, IN, USA, CAT# 4744934001). The optical density (OD) was read at 492 nm, and the blank OD was subtracted from all the OD values. The percentage of target cell lysis was calculated according to the following formula:$$Target\;cell\;lysis\left(\%\right)=\;\frac{\left[OD\;values\;of\;experimental\;release-OD\;value\;of\;TDCC\;background\right]}{\left[OD\;value\;of\;maximum\;release-OD\;value\;of\;minimum\;release\right]}\times 100\%$$

A four-parameter logistic regression was performed to fit the concentration–target cell lysis% curve using Prism7 (GraphPad Software). 

Human cytokine ELISA kits were used to quantify the production of interferon γ (IFN-ɣ, R&D Systems, VAL104C), tumor necrosis factor α (TNF-ɑ, R&D Systems, Minneapolis, MN, USA, CAT# VAL105G), IL-2 (Invitrogen, Thermo Fisher Scientific, Inc., Waltham, MA, USA, CAT# 88-7025-22), and IL-6 (R&D Systems, VAL102) in supernatants of the TDCC assay. During detection, the supernatants were diluted to appropriate concentrations as needed. The values were measured at 450 nm using a microplate reader (Spectramax M5; Molecular Devices, LLC., San Jose, CA, USA), and a four-parameter logistic regression was performed to fit the drug concentration–cytokine release curve using Prism7 (GraphPad Software).

To ensure the accuracy of the test results, the flow cytometry assay was used to assess the TDCC of BR109. As with the LDH assay above, we seeded 40,000 NCI-H929/100 μL/well into 96-well U-bottom plates, followed by the addition of 400,000 PBMCs/80 μL/well into 96-well U-bottom plates to make the effector cells (E): the tumor cells (T) = 10:1. Gradient concentrations of BR109 solution were prepared, and 20 μL/well of 10 × solution was added to the designated wells. At the same time, a positive control containing the mixture of 10 × anti-CD3 antibody (OKT3, eBioscience, Thermo Fisher Scientific, Inc., Waltham, MA, USA, CAT#16-0037-81), anti-CD28 antibody (CD28.2, Invitrogen, CAT#16-0289-85), and human IL-2 solution (Stemcell, Vancouver, BC, Canada, CAT#78036) was prepared and added at 20 μL/well to the corresponding wells so that the final concentration of CD3/CD28 antibodies was 1 μg/mL and the IL-2 solution was 10 ng/mL. The cells were cultured at 37 °C, 5% CO_2_ for 24 h. After the cells were washed, 5 µL of 7-AAD Viability Staining Solution (Biolegend, San Diego, CA, USA, CAT#420404) was added to obtain a final staining volume of 200 μL. The cells were gently vortexed and incubated for 15 min at room temperature in the dark. The samples could then be assessed using the BD Accuri™ C6 flow cytometer (BD Biosciences, Franklin Lakes, NJ, USA). 

### 2.7. In Vivo Efficacy Studies

For the MM model, NCI-H929 cells were collected while in the exponential growth phase and resuspended to a suitable concentration for subcutaneous tumor inoculation in NPG mice. Female NPG mice (Beijing Vitalstar Biotechnology Co., Ltd., Beijing, China) were used at 5 to 8 weeks of age; they had a weight of ~25 g. Each mouse was subcutaneously inoculated in the right flank with 5 × 10^6^ NCI-H929 cells, which had been resuspended in 0.1 mL of PBS and Matrigel (1:1). 

The evaluation of the efficacy of the engraftment of human PBMCs in NPG mice proceeded as follows. Briefly, 15 mice were divided randomly into 3 groups of 5 each. On the day of the NCI-H929 cell inoculation, 5 × 10^6^ PBMCs/0.1mL PBS (Milestone Biotechnologies, Shanghai, China) was injected intraperitoneally into each mouse. The inoculation day was defined as day 0. At day 0, sample collection and trial analysis were initiated. The effect of human immune cell reconstitution in human-PBMC-NPG mice was evaluated from three aspects: tumor cell proliferation, body weight and survival of the mice, and the proportion of each human immune cell subtype in the peripheral blood of the mice. The body weights and tumor volumes in the mice in all the groups were monitored at days 0, 5, 8, 12, 14, 16, 19, 21, 23, and 24. During the test period, the proportion of immune cells in the peripheral blood was determined by flow cytometry analysis at days 14 and 24.

Based on the best human PBMC reconstruction in the NCI-H929 xenograft NPG mouse models, the in vivo antitumor efficacy of BR109 could be evaluated. Similarly, on the day of the NCI-H929 cell inoculation, 5 × 10^6^ PBMCs/0.1mL PBS (Milestone Biotechnologies, Shanghai, China) was injected intraperitoneally into each mouse. When the tumor grew to a volume of approximately 80–150 mm^3^, all the mice were randomly grouped (*n* = 6 for each group). Unlike the above test, the day of grouping was defined as day 0. At day 0, the test drugs and control drugs were injected subcutaneously or intravenously. The treatments were administered twice per week for three weeks, and the observation periods could be extended as needed. The evaluation indicators of this experiment were tumor sizes and the health status of the test mice. The body weights for all the models and the tumor volumes were measured twice weekly. The tumor volume (TV) was measured and calculated using the formula $TV\left({mm}^{3}\right)=\frac{1}{2}\times length\times {width}^{2}$. The T/C value (T/C%) is the ratio of the average tumor weight in the treatment group (TmTV) to the average tumor weight in the control group (CmTV), and the tumor growth inhibition value was defined as TGI%. $T/C\%=\frac{TmTV}{CmTV}\times 100\%,$ $TGI\%=\left(1-\frac{T}{C}\right)\times 100\%$.

## 3. Results

### 3.1. Screening and Characterization of Anti-GPRC5D Antibody

Monoclonal antibodies against GPRC5D were obtained by immunizing fully humanized mice with human GPRC5D. Subcloned hybridoma cells were screened both positively and negatively. As shown in Figure 1a, 80A8, one of the positive clones, had similar species-binding properties to positive reference BM01, and they all clearly showed binding to hGPRC5D/cynoGPRC5D/mGPRC5D. Another clone, named 155C2, also showed a binding activity with hGPRC5D/cynoGPRC5D that was similar to that of 80A8, but the difference was that it did not bind to mGPRC5D. In terms of specificity, neither 80A8 nor 155C2 bound to human GPRC5A, GPRC5B, or GPRC5C (Figure 1b).

After sequencing, 80A8 and 155C2 were designed as recombinant fully human monoclonal antibodies of the IgG1 subclass that binds to hGPRC5D, and the binding activity is shown in Figure 1c. Apparently, the MFI values of 80A8 and 155C2 were significantly higher than those of BM02 upon the saturation of the binding. 

Binding competition assays were performed to determine the recognized similar epitopes of 80A8 and BM02. As shown in Appendix A, 80A8 and BM02 shared a large proportion of the epitopes, while 155C2 shared a small proportion with 80A8 and with BM02. Additionally, it was evident that 80A8 had a stronger ability to bind to hGPRC5D than BM02.

### 3.2. Construction and Stability Evaluation of Anti-GPRC5D × Anti-CD3 TCB

Both BR109 and BM03 had binding arms that could simultaneously recognize human GPRC5D and human CD3ɛ. As shown in Figure 2a, BR109 recognized HEK293T_hGPRC5D about two times better than positive reference BM03, with EC_50_ values of 3.207 nM and 7.079 nM, respectively, while negative reference IsotypeCD3 did not recognize HEK293T_hGPRC5D at all. Meanwhile, BR109 showed a slightly stronger binding activity than talquetamab analog (Figure 2b). The EC_50_ values were determined as 2.255 and 3.696 nM, respectively. When the bound TCBs reached a saturation plateau, 80A8 had higher MFI values than BM03 and talquetamab analog.

Accelerated stability tests at HT (40 °C) and different pH levels were frequently used to estimate the long-term stabilities of antibodies at a target storage temperature of 2–8 °C and in the different pH conditions of the purification process. Figure 2c,d showed the ability of BR109 to bind to HEK293T_hGPRC5D cells and Jurkat cells (with hCD3ɛ-binding site) under the accelerated conditions. It was evident that the treatment with HT and different pH conditions had little effect on its binding capacity (with similar EC_50_ values). Meanwhile, the purity of BR109 was verified by SEC. Compared with the pristine sample, the treated samples showed very weak differentiation (Appendix A).

### 3.3. Anti-GPRC5D × Anti-CD3 TCBs Can Induce T Cell Activation

In the in vitro T cell activation assay, the signal intensity detected by the microplate reader was directly proportional to the intensity of CD3 activation and signal transduction. As shown in Figure 3a,b, BR109 and positive control BM03 and talquetamab analog were all capable of inducing the activation of the Jurkat/NFAT-Luc cells 4E2 in a dose-dependent manner, while, with the negative control molecule in the presence of IsotypeCD3, no obvious activation signal was obtained. The BR109 molecule had the strongest specific activation effect, which was significantly better than that of BM03 and talquetamab analog, respectively. From the data point of view, the EC_50_ values of BR109 were three or two orders of magnitude lower than those of BM03 or talquetamab analog. Non-specific T cell activation was performed in the presence of BM03 or talquetamab analog with concentrations above 0.8 nM, but BR109 could specifically activate T cells only in the presence of MM tumor cells.

### 3.4. Anti-GPRC5D × Anti-CD3 TCBs Can Induce T-Cell-Mediated Cytotoxicity

As shown in Appendix A, the four cell lines MM.1R, MM.1S, NCI-H929, and RPMI 8226 all expressed human GPRC5D on the surface, and the expression of human GPRC5D was the highest in MM.1R, followed by MM.1S and NCI-H929, and the lowest was in RPMI 8226. The results of the T-cell-mediated cytotoxicity tested by LDH assay are shown, respectively, in Figure 4. Regardless of which of the MM.1R, MM.1S, and RPMI 8226 cell lines was used, the anti-GPRC5D × anti-CD3 TCB was able to induce concentration-dependent cell lysis. Among them, the killing activity of PBMCs (donor ID: LP221213007) induced by BR109 was significantly better than that of the positive control BM03 and talquetamab analog. Further confirmation experiments demonstrated the specific killing of the MM cell line NCI-H929 by three different human PBMC donors. The data were detected using the flow cytometry method. It is shown in Appendix A that BR109 showed good TDCC, which was stimulated by three different human PBMC donors, and it was concentration-dependent. These data demonstrated the excellent antitumor activity of BR109 in vitro.

Moreover, to provide evidence that anti-GPRC5D × anti-CD 3 TCBs triggered T cell activation, but those others did not, the levels of the secreted cytokines in the supernatants of BR109 or IsotypeCD3 were added to cocultures of GPRC5D-expressing MM.1R or MM.1S cells and human PBMCs (donor ID: LP221213007) during the TDCC tests. Compared with negative control IsotypeCD3, BR109 treatment led to the obvious and concentration-dependent secretion of TNF-α, IFN-γ, IL-2, and IL-6, consistent with T cell activation and T-cell-mediated cytotoxicity (Figure 5).

### 3.5. The Antitumor Efficacy of Anti-GPRC5D × Anti-CD3 TCBs in the NCI-H929 Xenograft NPG Mouse Models

First, in the NCI-H929 xenograft NPG mouse models with human immune cell reconstitution (donor ID: DZ211314), BR109 completely prevented tumor formation at a 1 mg/kg dose (Figure 6). As shown in Appendix A, at day 18, the mean tumor volume (mTV) of the vehicle control group was 3303.18 mm^3^; the mTVs of the BM03 and BR109 groups were 2429.01 mm^3^ and 85.62 mm^3^, respectively, and their TGIs were 26.46% (*p* = 0.087) and 97.41% (*p* < 0.001). The tumor inhibition rate of BR109 was significantly better than that of blank control PBS and positive reference BM03. At day 18, after the last dosing of the test compounds, the observation was extended for 9 days. According to Appendix A and Appendix A, tumor recurrence was not found during the study period. Meanwhile, no mice were found to have died during the process, and the lowest body weight change rate was −2.26% (BM03 group). This suggests that all the study drugs were generally well-tolerated.

Using PBMCs from three unrelated donors, we evaluated the efficacy of human immune cell reconstitution in the NCI-H929 xenograft NPG mouse models. As shown in Appendix A, the NCI-H929 tumor-bearing mice were successfully inoculated with three human PBMCs, and the mice remained active and in a condition of good health during the experiment. The percentage of human immune cells at day 24 was significantly higher than that at day 14 (Appendix A). Furthermore, at day 24, the proportion of human immune cells (human CD45^+^, CD3^+^, CD4^+^, and CD8^+^ cells) in donor DS20219 and donor DB21216 was significantly higher than that in donor NF0055 (Appendix A). Among these, the ratios of human CD3^+^ cells and human CD8^+^ cells in donor DB21216 were higher than those in donor DS20219.

To further test the antitumor effects, three doses of BR109 administration in gradients (0.8, 0.2, and 0.05 mg/kg) were injected into the NCI-H929 xenograft NPG mouse models with human immune cell reconstitution (donor ID: DB21216). As shown in Figure 7 and Appendix A, when compared with negative control IsotypeCD3, significant reductions in tumor volumes were observed at all the doses of BR109 delivered. Compared with the positive control talquetamab analog, BR109 had stronger tumor-suppressive activity; especially at the dose of 0.05 mg/kg in particular, the tumors were essentially eliminated. At the doses of 0.8 and 0.2 mg/kg, BR109 could eliminate tumors completely, especially at the dose of 0.2 mg/kg, and it was determined that both subcutaneous (s.c.) administration and intravenous (i.v.) administration could have this effect. According to Appendix A and Appendix A, no mice were found to have died during the process, and the lowest body weight change rate was −0.86% (BR109 0.8mg/kg i.v. group). Overall, the test drugs were found to be safe and well-tolerated. 

## 4. Discussion

Previous studies found that the expression of GPRC5D protein on the tumor cells of MM patients was specific, and, as a seven-transmembrane protein, it was difficult to detach, leading to off-target effects of TCBs [15,19]. Notably, GPRC5D was a potential target for MM treatment. Therefore, the discovery of antibodies against the human GPRC5D protein was the top priority in establishing this antibody therapy. Monoclonal antibodies obtained from hybridomas in Balb/c wild-type mice still needed complex humanization to eliminate their potential immunogenicity. The antibodies produced from the fully humanized genetically engineered mice did not require humanization and affinity maturation and maintained the absolute advantage of the natural binding affinity.

On the other hand, as GPRC5D is a type of transmembrane protein with a complex structure, we specifically used the flow cytometry method for screening cell binding, instead of using the GPRC5D protein, to ensure that the screened antibodies could recognize the real three-dimensional structures of GRPC5D on the membrane surface. The positive clone, 80A8, that we screened bound specifically to hGPRC5D/cynoGPRC5D/mGPRC5D but did not bind to human GPRC5A, GPRC5B, or GPRC5C. Monoclonal antibody 80A8 showed higher binding activity than positive control BM02 (the variable region sequences were obtained from talquetamab). Compared with another positive clone, 155C2, which was screened at the same time, 80A8 had a GPRC5D-binding epitope more like the anti-GPRC5D arm of talquetamab (Appendix A). Although 80A8 and 155C2 had similar binding activity (Figure 1c), we eventually selected 80A8, which has binding activities in three species: human/cynomolgus monkey/mouse.

We combined 80A8 with human CD3 antibody SP34 to construct the TCB BR109 in an asymmetric format, “scFv-Fab-Fc”. BR109 had similar architecture to the natural antibodies, with less immunogenicity and a longer half-life. The KiH mutations described by Carter and colleagues were used in the constant region of the antibody (Knob: T366W; Hole: T366S, L368A, and Y407V) [30] and could reduce the homodimeric byproduct in the process of antibody expression. Fc-mediated antibody-dependent cell-mediated toxicity (ADCC), antibody-dependent cell-mediated phagocytosis (ADCP), and complement-mediated cytotoxicity were reduced by the LALA mutations (L234A and L235A) in the Fc portion [31]; all of them sometimes resulted in cytokine storms, in which numerous inflammatory cytokines from NK and other immune cells were secreted abnormally [32]. 

In this study, purified TCB BR109 underwent accelerated treatment at high temperature (40 °C) and under different pH conditions. It was found that different treatment conditions had almost no impact on the antigen-binding activity and purity of BR109, reflecting its excellent stability. The in vitro antitumor activity of BR109 was also evaluated in this study. One positive control, talquetamab analog, had the same structure and sequences as talquetamab, which was from J&J and has been commercially available. Moreover, another positive control, BM03, a TCB molecule with the same structure as BR109 and the same sequences as talquetamab, was also used. In MM cell line MM.1R, BR109 showed a higher ability to activate T cells and was significantly better than talquetamab analog and BM03. The study further determined the T-cell-mediated cytotoxicity of BR109 in three MM cell lines with different GPRC5D expression levels under random human PBMC donors. Since LALA mutations were introduced to avoid the GPRC5D-independent activity caused by Fcγ receptors in TCBs and CD3-expressing cells [33,34], PBMCs, which are able to simulate the immune environment in human blood circulation with a realistic proportion of immune cells, could be used to evaluate the specific cytotoxicity of TCB in vitro in methodological aspects. According to the MM cell killing status of the negative control IsotypeCD3 in each test, all three TCBs in the two MM cell lines, MM.1R and MM.1S, were able to specifically induce T-cell-mediated cytotoxicity except for RPMI 8226, which had obvious non-specific T cell activation in the presence of high concentrations of TCBs. Overall, the T-cell-mediated MM cell lines’ killing rate of BR109 was almost the same as that of positive controls talquetamab analog and BM03, but the EC_50_ value was 1–2 orders of magnitude lower than that of talquetamab analog and BM03, indicating that BR109 has stronger T-cell-mediated cytotoxicity. In addition, human PBMC from three different donors and MM cell models were used to verify the in vitro activity. The results confirmed that BR109 had concentration-dependent T-cell-mediated cytotoxicity in MM cell line NCI-H929 and that different donor PBMCs could be stimulated to exert specific antitumor activity. At the same time, cytokine release experiments could prove that BR109 was able to specifically activate T cells to exert cytotoxicity and also enabled a comparison with negative control IsotypeCD3. Taken together, BR109 showed excellent in vitro performance activity.

The six treatments with 1 mg/kg BR109 Q2W resulted in profound tumor regression in the established human-immune-reconstituted NCI-H929 NPG mouse xenograft model without adverse outcomes (e.g., mouse death, severe weight loss, etc.). The antitumor performance of BR109 was significantly better than that of positive control BM03. The studies above were able to prove the excellent properties of anti-GPRC5D antibody 80A8 that we screened. To rule out the possibility of tumor elimination caused by PBMC-mediated non-specific cytotoxicity in these models, we selected human PBMCs from three different donors. In the above mouse models without BR109 administration, the NCI-H929 cells inoculated subcutaneously could proliferate normally, and no adverse consequences, such as mouse death and severe weight loss, were found during the 3-week observation period. In summary, severe graft-versus-tumor (GVT) and graft-versus-host-disease (GVHD) effects were not observed in these models. Among the three random donors, we selected the one with the best human immune cell reconstruction effect to verify that the antitumor efficacy of BR109 in vivo was not accidental. In this study, we refined the dosing regimen, setting talquetamab analog as a positive control. Very significantly, 0.05 mg/kg BR109 showed an excellent antitumor effect after two doses, which was better than the efficacy of talquetamab analog at the same dose; in particular, it was slightly better than the talquetamab analog efficacy at 0.2 mg/kg. No adverse results (such as mouse death or severe weight loss) were found throughout the trial. The above experiments further prove the excellent antitumor activity of anti-GPRC5D × anti-CD3 TCB BR109. Meanwhile, the antitumor effects of the subcutaneous and intravenous administration of BR109 at a dose of 0.2 mg/kg were almost the same. This suggests that priority could be given to subcutaneous administration as the clinical route of administration in the future, with less severe side effects and more convenient operation.

In the future, efficacy trials can be conducted in other human-immune-reconstituted MM cell NPG mouse xenograft models (e.g., MM.1R and MM.1S) by setting lower dosages or extending the dosage interval (e.g., QW) to obtain more abundant in vivo potency data for BR109. As the binding affinity of 80A8 against cynoGPRC5D was one order of magnitude lower than that of hGPRC5D (Appendix A), we did not conduct toxicological studies related to BR109 in cynomolgus monkeys. Future trials will use GPRC5D/CD3 double-humanized mouse models for safety and efficacy evaluation. The current safety issues related to anti-GPRC5D × anti-CD3 TCB could be evaluated using the clinical data obtained from talquetamab and RG6234 [22,24,35].

## 5. Conclusions

In this study, a fully human mAb 80A8 against highly specific MM target GPRC5D was developed rapidly and was reliably based on humanized mice. Based on 80A8, we constructed anti-GRPC5D × anti-CD3 TCB BR109 with higher affinity activity and antitumor activity than positive references talquetamab analog and BM03. These data support further development of BR109 for the clinical treatment of relapsed and refractory MM.

## 6. Patents

Patent applications related to this work have been filed by BioRay Co., Ltd., Taizhou, China (Application No. 202310772444.6).

## Figures and Tables

**Figure 1 cancers-15-05774-f001:**
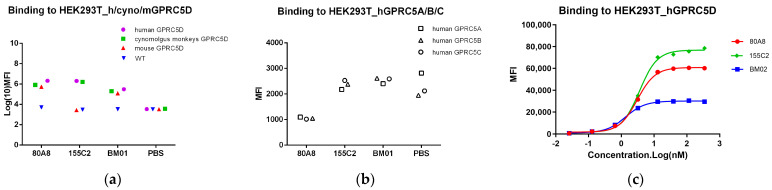
The binding activities of anti-GPRC5D antibodies. (**a**) The binding activities of antibodies produced by hybridoma with HEK293T cells expressing GPRC5D of different species were determined by flow cytometry assay and were plotted as dots. BM01 was a positive control. WT: wild-type HEK293T cells. (**b**) The binding activities of antibodies produced by hybridoma with HEK293T cells expressing human GPRC5A, GPRC5B, and GPRC5C were determined by flow cytometry assay and were plotted as dots. (**c**) The binding activities of fully human monoclonal antibodies 80A8, 155C2, and BM02 with HEK293T_hGPRC5D were graphed using MFI.

**Figure 2 cancers-15-05774-f002:**
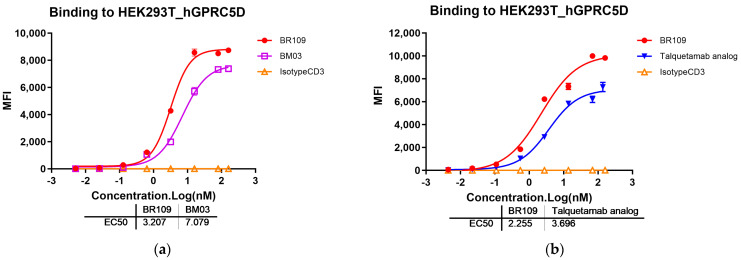
The binding activities of anti-GPRC5D × anti-CD3 TCBs with HEK293T_hGPRC5D were determined by flow cytometry assay and were plotted as MFI. (**a**) BR109, negative control IsotypeCD3, and positive control BM03; (**b**) BR109, negative control IsotypeCD3, and positive control talquetamab analog. The binding activities of BR109 with various accelerated treatment conditions with human CD3ε protein and HEK293T_hGPRC5D cells; (**c**) was graphed using OD_450nm_; (**d**) was graphed using MFI. Data are shown as mean ± SD (*n* = 3/group).

**Figure 3 cancers-15-05774-f003:**
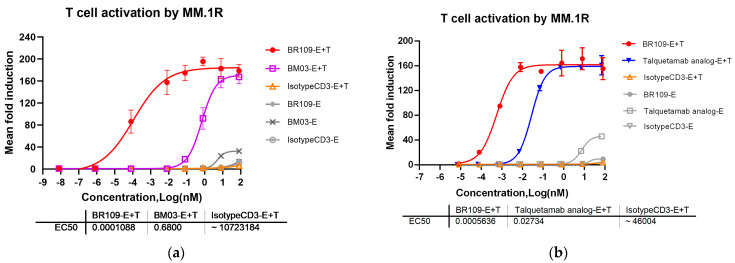
T cell activation of anti-GPRC5D × anti-CD3 TCBs after coculturing with Jurkat/NFAT-Luc cell line 4E2 (effector cells) and multiple myeloma cell line MM.1R (tumor cells). (**a**) Contained BR109, positive control BM03, and negative control IsotypeCD3; (**b**) contained BR109, positive control talquetamab analog, and negative control IsotypeCD3. “E” stands for effector cells and “T” stands for tumor cells. Colored lines represent the experimental system containing effector cells and tumor cells, while gray lines represent containing effector cells only. Data are shown as mean ± SD (*n* = 3/group).

**Figure 4 cancers-15-05774-f004:**
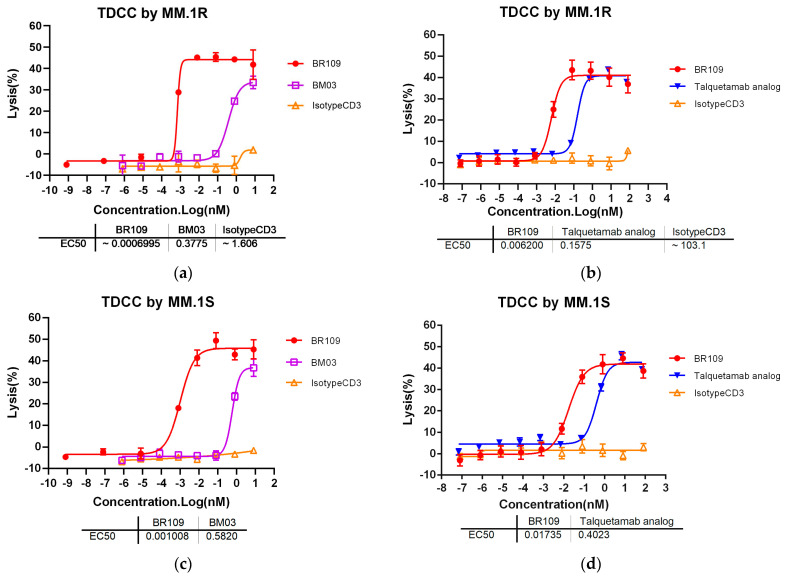
Cytotoxicity of anti-GPRC5D × anti-CD3 TCBs when coculturing multiple myeloma cell lines with human PBMCs (donor ID: LP221213007). (**a**) BR109, positive control BM03, and negative control IsotypeCD3-mediated TDCC by MM.1R; (**b**) BR109, positive control talquetamab analog, and negative control IsotypeCD3-mediated TDCC by MM.1R; (**c**) BR109, positive control BM03, and negative control IsotypeCD3-mediated TDCC by MM.1S; (**d**) BR109, positive control talquetamab analog, and negative control IsotypeCD3-mediated TDCC by MM.1S; (**e**) BR109, positive control BM03, and negative control IsotypeCD3-mediated TDCC by RPMI 8226; (**f**) BR109, positive control talquetamab analog, and negative control IsotypeCD3-mediated TDCC by RPMI 8226. Data are shown as mean ± SD (*n* = 3/group).

**Figure 5 cancers-15-05774-f005:**
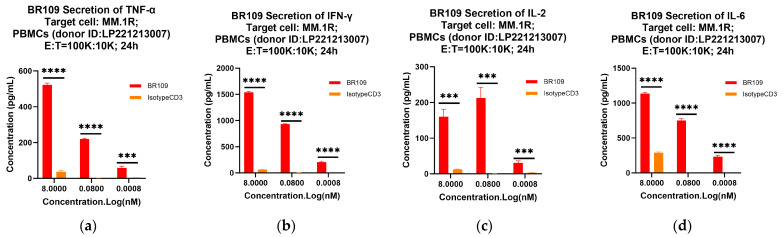
Secretion of cytokines by anti-GPRC5D × anti-CD3 TCBs when coculturing multiple myeloma cell lines with human PBMCs (donor ID: LP221213007). Secreted cytokines in the supernatants after 24 h culture was quantified using TNF-α, IFN-γ, IL-2, and IL-6 Cytokine Kit. (**a**–**d**) show the secretion of four cytokines, respectively, in the presence of MM.1R; (**e**–**h**) show the secretion of four cytokines, respectively, in the presence of MM.1S. Data are shown as mean ± SD (*n* = 3/group). Parametric Dunnett test: ***, *p* < 0.001; ****, *p* < 0.0001.

**Figure 6 cancers-15-05774-f006:**
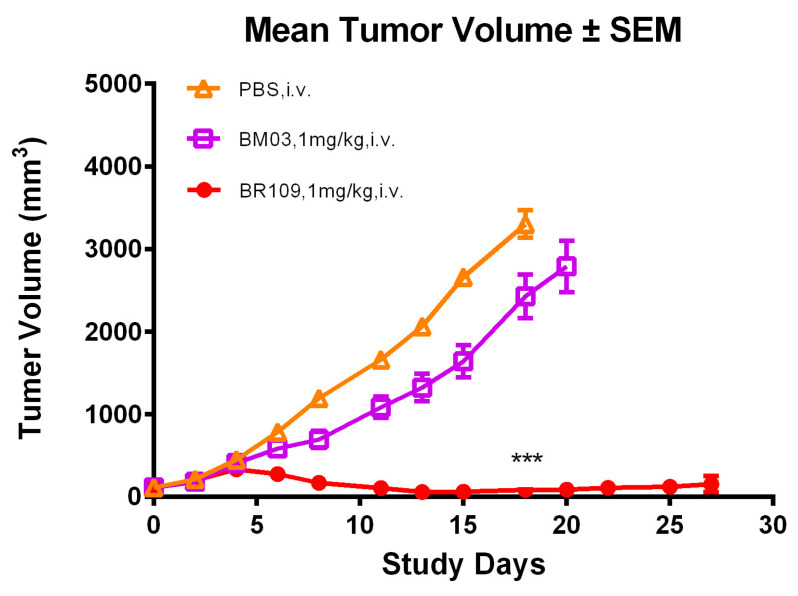
Antitumor activity of BR109 and positive control BM03 against multiple myeloma cell line H929 in NPG mouse models with human PBMCs (donor ID: DZ211314). Parametric Dunnett test: *** *p* < 0.001, versus PBS treatment at day 18.

**Figure 7 cancers-15-05774-f007:**
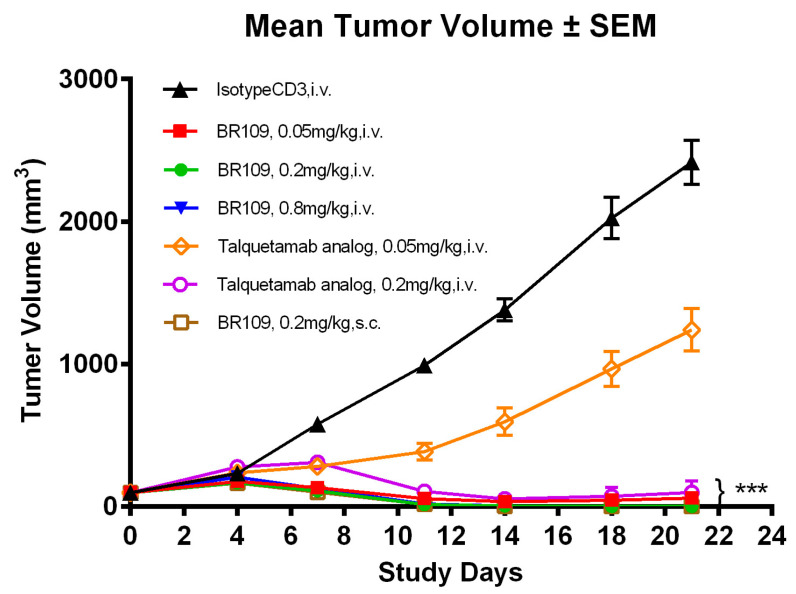
Antitumor activity of BR109 and positive control talquetamab analog against multiple myeloma cell line H929 in NPG mouse models with human PBMCs (donor ID: DB21216). Parametric Dunnett test: *** *p* < 0.001, versus IsotypeCD3 treatment at day 21.

## Data Availability

All data in the current study are available from the corresponding author upon reasonable request.

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
