# Peer review of "BR109, a Novel Fully Humanized T-Cell-Engaging Bispecific Antibody with GPRC5D Binding, Has Potent Antitumor Activities in Multiple Myeloma"

_cancers, 2023, doi:10.3390/cancers15245774_

Round 1
Reviewer 1 Report
Comments and Suggestions for Authors
This is a very interesting study, I have some questions/comments:
1. Line 40 - MM is a kind of hematological malignancies - please correct
2. Lines 41,42 - please use correct definition of CRAB symptoms
3. Line 42 - extensive bone destruction, pathological fractures, osteolytic lesions, and bone loss - please shorten
4. Line 60- GPRC5D mRNA levels are higher in MM plasma cells than normal cells - what normal cells?
5. Line 67- to some available statistics - what does that mean?
Comments on the Quality of English Language
I think a proofreading by a native speaker would be advisable.
Reviewer 2 Report
Comments and Suggestions for Authors
Review of paper by Liu et al. for Cancers on 11-16-23
Comments for Authors
Line 40- The word “Myeloma” should not be capitalized.
Line 71- The word “Talquetamab” should also not be capitalized throughout the paper unless it is the first word in the sentence.
Line 72- The route of administration has nothing to do with the delay in therapy with CAR T-cell therapies. It is because of the time it takes to manufacture these modified cells.
Line 242- The word “human” should be added before the “immune.”
Line 249- How was the best human PBMC reconstruction determined?
Line 322- The end of the sentence should be ”analog, respectively.”
Figure 3- 4E2 is not defined in the figure. Are these the effector cells?
Lines 339-340- What is meant by the term “concentration dependent crack?” The authors use the term “killing activity of T cells” but the proper term is “killing activity of PBMCs” because these are not purified T cells.
Figure 4- What are the number of PBMCs in these experiments?
Line 360- “Firstly” should be “First.”
Line 361- Define “human immune cells.”
Line 386- Define the term “more gradients of BR109?”
Line 387- Which MM xenograft was studied?
Lines 389-391- This needs to be rewritten. It is confusing as stated.
Line 390- This is a dose NOT a concentration.
Lines 391-393- This was evaluated at only 1 dose so conclusions regarding the relative efficacy of these two modes of administration is not possible.
Line 413- This should be “is” NOT “was.”
Why did the authors not also study fresh MM bone marrow for the in vitro studies?
The MM.1 models are exquisitely sensitive to the anti-MM effects of all anti-MM drugs. The study would be enhanced by using other MM cell lines to conduct their experiments.
Comments on the Quality of English Language
Minor editing of English language required
Reviewer 3 Report
Comments and Suggestions for Authors
The authors proposed a new anti-GPRC5D antibody and the construction of an anti-GPRC5D×anti-CD3 TCB. They tested the antibody both in vitro and in vivo.
The paper is well written and organized. However, I have some specific issues.
1- Figure3: the authors performed the T cell activation assay using a T cell line. I would suggest the authors to use human T cells (from healthy donors) and to check The T cell activation by analyzing the expression of CD69 and CD25 by flow cytometry. moreover, the analysis of specific cytokines in the supernatant of the cocolture should be performed.
2- The use of human T cells, instead of PBMC, is also suggested for T-cell mediated cytotoxicity (Figure 4). Which is the contribution of NK cells mediated cytotoxicity using PBMC?
Moreover, mononuclear cells isolated from MM patients should be tested to verify the efficacy of the antibody in an ex vivo experiment
Round 2
Reviewer 3 Report
Comments and Suggestions for Authors
Figure 5: there is no statistic in the graph. Please add the standard deviation and p-value. How many replicates?
Comment 1: I agree with the authors that the analysis of CD69 and CD25 is a classic method but it is the only one that allow the phenotipic characterization of T cell activation.
Comments on the Quality of English Languagegood
